# Elevated NETs and Calprotectin Levels after ChAdOx1 nCoV-19 Vaccination Correlate with the Severity of Side Effects

**DOI:** 10.3390/vaccines10081267

**Published:** 2022-08-06

**Authors:** Geir Hetland, Magne Kristoffer Fagerhol, Markus Karl Hermann Wiedmann, Arne Vasli Lund Søraas, Mohammad Reza Mirlashari, Lise Sofie Haug Nissen-Meyer, Mette Stausland Istre, Pål Andre Holme, Nina Haagenrud Schultz

**Affiliations:** 1Department of Immunology and Transfusion Medicine, Oslo University Hospital Ullevål, 0450 Oslo, Norway; 2Institute of Clinical Medicine, University of Oslo, 0372 Oslo, Norway; 3Department of Neurosurgery, Oslo University Hospital Rikshospitalet, 0424 Oslo, Norway; 4Department of Microbiology, Oslo University Hospital Rikshospitalet, 0424 Oslo, Norway; 5Department of Haematology, Oslo University Hospital Rikshospitalet, 0424 Oslo, Norway; 6Department of Haematology, Akershus University Hospital, 1478 Lørenskog, Norway

**Keywords:** adverse drug event, calprotectin, COVID-19 vaccines, NETs, treatment outcome

## Abstract

ChAdOx1 nCoV-19 vaccination has been associated with the rare side effect; vaccine-induced immune thrombotic thrombocytopenia (VITT). The mechanism of thrombosis in VITT is associated with high levels of neutrophil extracellular traps (NETs). The present study examines whether key markers for NETosis, such as H3-NETs and calprotectin, as well as syndecan-1 for endotheliopathy, can be used as prognostic factors to predict the severity of complications associated with ChAdOx1 vaccination. Five patients with VITT, 10 with prolonged symptoms and cutaneous hemorrhages but without VITT, and 15 with only brief and mild symptoms after the vaccination were examined. Levels of H3-NETs and calprotectin in the vaccinated individuals were markedly increased in VITT patients compared to vaccinees with milder vaccination-associated symptoms, and a strong correlation (r ≥ 0.745, *p* < 0.001) was found with severity of vaccination side effects. Syndecan-1 levels were also positively correlated (r = 0.590, *p* < 0.001) in vaccinees to side effects after ChAdOx1 nCoV-19 vaccination. We hypothesize that the inflammatory markers NETs and calprotectin may be used as confirmatory tests in diagnosing VITT.

## 1. Introduction

Vaccine-induced immune thrombotic thrombocytopenia (VITT) is a rare side effect of adenoviral vector vaccines associated with high mortality [1]. It was first reported in March 2021 when five cases at Oslo University Hospital were admitted with severe thrombosis combined with low platelet counts 7–10 days after vaccination with 1st dose of ChAdOx1 nCoV-19 [2]. Similarly, in Germany and Austria 11 cases were reported with thrombosis and trombocytopenia after ChAdOx1 nCoV-19 vaccination [3], and later several cases have been reported [4]. VITT has also been observed after the Ad26.COV2.S [5], but the incidence seems to be lower than for the ChAdOx1 nCov-19 Vaccine [6]. The discovery of VITT has impacted on the vaccination strategy in several countries, and the knowledge about VITT syndrome, its symptoms and treatment has increased the survival rate [1]. VITT is a distinct entity of thrombotic thrombocytopenia and treatment with therapeutic doses of anticoagulation in combination with intravenous immunoglobulins has been shown to improve the outcome [1].

Patients with VITT have high levels of antibodies to platelet factor 4 (PF4)–polyanion complexes [2] with platelet activating ability, and the mechanism of thrombosis and bleeding resembles that of autoimmune heparin-induced thrombocytopenia [3]. Recently, it has been demonstrated that the thrombus generation in the VITT patients was accompanied by a fulminant neutrophil activation and formation of neutrophil extra cellular traps (NETs), i.e., NETosis [7].

In April 2021 over 200 thrombosis cases, ¾ being cerebral, had been reported to EudraVigilance amongst 34 million people vaccinated in the EU and UK with the ChAdOx1 nCov-19 Vaccine [8]. In an observational study among 500 Norwegian health workers vaccinated with ChAdOx1 nCoV-19, the majority had normal platelet counts and negative screening for antibodies to anti-PF4–polyanion antibodies [9]. The discovery of VITT has created concerns and early recognition of the syndrome was deemed important to improve the outcome [1]. In fact, in an updated EudraVigilance report as of December 2021 from EU/EEA countries, there were over 220,000 cases of suspected side effects with COVID-19 Vaccine Vaxzevria (AstraZeneca, Luton, United Kingdom) (ChAdOx1 nCoV-19 vaccine) after the administration of nearly 69 million doses. Among these 1300 had fatal outcomes [10]. Pre-VITT syndrome has also been described, which is characterized by thrombocytopenia and severe headache without associated thrombosis following vaccination with the ChAdOx1 nCoV-19 vaccine [11]. 

Calprotectin, NETs and syndecans are key inflammatory markers with physiological effects, involved in innate immune activation and vascular endothelial damage, respectively. Calprotectin is the main cytosolic protein in neutrophil granulocytes [12]. It is used as a sensitive fecal marker in gastrointestinal diseases [13], and has antimicrobial and cytotoxic properties through its zinc-binding activity [14,15]. Calprotectin is associated with severe pulmonary disease in COVID-19 [16], and may help discriminate severe from mild COVID-19 [17].

NETs are webs of DNA, surrounded by histones and granular proteins that are expulsed from neutrophils for trapping and destruction of invading pathogens in the extracellular space. Elevated levels of NETs have been demonstrated in hospitalized patients with COVID-19 [18], and NETosis has also been associated with thrombosis formation in patients with VITT [4,19]. Circulating NETs are removed from blood by extracellular DNAses that digest free DNA [20,21].

Syndecan-1 is a glycocalyx marker for damaged endothelium in the vasculature. Vascular endothelial damage and dysfunction is suggested as prognostic indicator for COVID-19 [22], In this disease there is a direct effect of immune response dysregulation on endothelial damage, which is an essential pathological response to the infection that in severe cases results in respiratory and multi-organ failure [23]. The endotheliopathy can also promote thrombosis [23], and thrombosis profiling may reveal endothelial damage in COVID-19 patients [24]. Therefore, not surprisingly, elevated levels of syndecan-1 have been demonstrated in COVID-19 disease [24].

We wanted to investigate whether key inflammatory markers related to NETosis and endothelial damage are increased after ChAdOx1 nCoV-19 vaccination and whether their levels are associated with the severity of side effects from the vaccination. Since there are few previous reports on these inflammatory markers after ChAdOx1 vaccination for COVID-19 relative to clinical outcome, their examination in this context may contribute to the understanding of the serious VITT side effects.

## 2. Materials and Methods

### 2.1. Patients

We examined blood samples from individuals after vaccination with the ChAdOx1 nCov-19 vaccine. The individuals were divided into three groups according to different degrees of vaccine complications. In addition, we included a control group consisting of healthy, non-COVID-19 vaccinated blood donors, sampled in 2015. Group 1 consisted of patients with VITT previously described.^2^ Onset of symptoms in this group was 7–10 days after vaccination. Group 2 consisted of individuals who had contacted health services with prolonged symptoms (mild to strong headaches) and signs (cutaneous hemorrhages) with median onset of 1.5 days after vaccination and duration of 2.8 weeks. Group 3 consisted of individuals with none or brief, mild symptoms (mild headaches) with an onset 0–2 days after vaccination (Figure 1). 

Vaccination side effects score was defined as: 3 for VITT, 2 for prolonged symptoms and signs, 1 for brief mild symptoms, and 0 for non-vaccinated blood donor controls.

### 2.2. Blood Collection and Processing

Samples from VITT and prolonged symptoms patients were sera and the other samples were EDTA plasma. Whereas samples from the vaccinees were collected within 7–64 days after receiving the ChAdOx1 nCov-19 vaccine, samples from healthy blood donors were collected in 2015, before the pandemic. All samples were kept frozen at −80 °C until analysis. 

### 2.3. NETs Assays

Assay for NETs by Histone 3–calprotectin hybrid ELISA was performed as described in Fagerhol et al., 2020) [25]. Briefly, rabbit anti-Histone was coated in Nunc Maxisorp microwells (Sigma-Aldrich, Darmstadt, Germany). To obtain a standard curve, separate wells were coated with a mixture of monoclonal (=MiMo) anti-calprotectin antibodies from six different clones. After incubation of samples and standards (recombinant calprotectin at 5 to 500 ng/mL) for one hour, wells were incubated for 40 min with horse radish peroxidase (HRP) conjugated MiMo. Readings were performed after 20–30 min with substrate.

### 2.4. Calprotectin Mixed Monoclonal Assay

A novel calprotectin ELISA based on a mixture of monoclonal antibodies was established [26]. Since a considerable proportion of calprotectin in biological materials contain both histone and DNA fragments, a combination of several, different monoclonals will be necessary to obtain a reliable assay. The monoclonals here were selected so that the mixture reacted with all fractions obtained from chromatography of stool extracts from patients with inflammatory bowel syndrome. The mixed monoclonal (MiMo) antibodies were used both for coating of microwells and preparation of a HRP-conjugate. The assay range of the ELISA was from 5 to 1000 ng/mL with a CV of about ten per cent for the standards.

### 2.5. Syndecan-1 Assay

Serum levels of syndecan-1 were determined using a commercial ELISA kit detecting human syndecan-1 (Diaclone, 950.640.192, Besançon, France). Values are given as ng/mL.

### 2.6. Statistics

One way ANOVA was used for examining differences between several groups either on means when data was normally distributed, or on ranks when not. Given significant differences between the groups in ANOVA, differences between two and two groups were examined by *t*-test (parametric data) or Wilcoxon rank sum test (non-parametric data). Correlation between groups was examined similarly, by Pearson correlation for parametric data, otherwise by Spearman’s correlation. As statistical/graph package, GraphPad Prism 8.4.3 (GraphPad Software, San Diego, CA, USA) was employed. *p* values < 0.05 were considered significant.

## 3. Results

### 3.1. Patient Characteristics

We examined samples from 5 patients with VITT (group 1), 10 individuals with strong prolonged symptoms (group 2), 15 patients with no/brief symptoms (group 3) and 20 healthy controls (group 4). Characteristics of the study participants are summarized in Table 1 and shown individually in Table 2. The ages in the groups were similar (*p* = 0.200). However, there was an overweight of women in the vaccinated groups, varying from 80–100% (Table 1). Whereas platelet counts were normal and similar for patient groups 3 and 2 (*p* = ns), they were 10-fold lower in those with VITT (group 1) (*p* < 0.0001). In fact, there was a strong negative correlation (r = −0.763, *p* < 0.0001) between symptoms severity and platelet count (Table 3). 

### 3.2. NETs, Calprotectin and Syndecan-1 Levels Are Increased

Analysis of H3-NETs showed a significant difference between the groups (One way ANOVA *p* < 0.0001). Group 1 had 6-fold* higher NETs levels than group 2, which again had over twice as high NETs levels as those in group 3 (*p* < 0.0001). These also tended to have higher NETs values than group 4 (*p* = 0.088) (Figure 2). 

For calprotectin values there was also significant difference between the groups (*p* < 0.0001). Calprotectin levels in group 1 were 2-fold and near 3-fold higher compared with those in group 2 and group 3, respectively (*p* < 0.0001). The latter had further over 10 times higher levels than group 4 (*p* < 0.0001) (Figure 3).

When syndecan-1 was analyzed in the ChAdOx1 nCoV-19 vaccinated individuals and blood donors, the values were found to be significantly different between the groups (*p* = 0.0001). However, whereas syndecan-1 levels in group 1 were over 2-fold higher than those in group 2 (*p* = 0.023), there was no difference between group 2 and 3 (Figure 4). On the other hand, the latter group had 3-fold higher syndecan-1 levels compared with group 4 (p=0.017), which also was true for the group 2 (*p* = 0.005) (Figure 4). 

### 3.3. Correlations

Importantly, when the levels of the different markers were related to side effect severity graded 3–1 in the vaccine groups and with 0 for non-vaccinated healthy controls, there were high positive correlations between levels of calprotectin (r = 0.902), NETs (r = 0.745), and to a lesser degree syndecan-1 (r = 0.590) on one side, and side effects on the other side (Table 3). When examining the inter-relationship between levels of the above inflammatory markers, H3-NETs was found to correlate highly positively with calprotectin (r = 0.818, *p* < 0.0001), and to a lesser degree with syndecan-1 (r = 0.427, *p* = 0.018) (Table 3). However, there was no correlation between calprotectin and syndecan-1 (r = 0.189, ns) (Table 3). Platelet counts correlated negatively with levels of both NETs (r = −0.813), calprotectin (r = −0.626) and syndecan-1 (r = −0.488) (Table 3).

## 4. Discussion

We have investigated whether key inflammatory markers related to NETosis and endothelial damage are increased after ChAdOx1 nCoV-19 vaccination, and whether there is an association between levels of these markers and severity of vaccination side effects.

The main finding of this study is that severe side effects (VITT) are associated with high levels of NETs. Also, measurements of calprotectin, which is contained in NETs [27] together with DNA histone and granular enzymes, as well as syndecan-1, did discriminate between patients with VITT and those with prolonged symptoms and signs but without having VITT. Recently, increased NET formation, as measured by citrullinated H3, and also higher dsDNA levels was described in VITT patients. Furthermore, it has already been shown that that there are increased levels of neutrophil intracellular proteins S100A8/A9 (calprotectin) in group 1 [7]. In the present study, however, we demonstrated that the NETs levels correlate with the severity of side effects of the ChAdOx1-vaccine. Further, whereas both H3-NETs and calprotectin levels did discriminate between the prolonged severe and the mild symptoms groups 2 and 3 (Figure 1), syndecan-1 levels did not. Calprotectin and syndecan-1 were also increased in group 3 (individuals with brief symptoms) compared with the control group (healthy blood donors), but this was not observed for NETs. 

Hence, if raised NETs levels can be detected early on after ChAdOx1-vaccination, this marker may be a predictor for a severe outcome of vaccine complications. To monitor possible emerging side effects, NETs analysis could also be used after vaccination with the COVID-19 Janssen vaccine, which also is an adenoviral vector vaccine with the potential to trigger VITT [6]. The raised levels of syndecan-1 in individuals with brief mild or prolonged symptoms after such vaccination and the 2-fold levels increase in syndecan-1 in the VITT patients, suggests that some initial endotheliopathy has developed [23], which increases significantly in the VITT group. Whether this further promotes thrombosis resulting in VITT is possible but yet not known. As expected, there was a strong negative correlation between severity of symptoms and platelet counts in the vaccinees. Similarly, there were also negative correlations between levels of the inflammatory markers and platelet counts.

This study has several limitations. First, the number of study participants with different categories of side effects is limited and impacts statistical power. The categorization itself may be criticized, as side effects vary between vaccinees, leading to selection bias. However, consistent patterns in the clinical presentation of side effects after vaccination were observed, leading to a categorization into groups 1, 2 and 3. 

One strength of the study is the comparison of three patient groups with varying symptoms and outcome after ChAdOx1-nCoV19 vaccination with non-vaccinated healthy controls from a similar population, as many health workers are also blood donors, who also were from the pre-COVID-19 era and thus had not encountered the SARS-CoV-2 virus. Another strength is the examination of highly relevant assays for inflammation, as demonstrated by the high degree of correlation between their results and the degree of side effects severity after the vaccination.

Side effects after ChAdOx1-nCoV19 vaccination were mainly observed in women in the Norwegian population, indicating an association with gender. However, the vaccine was primarily distributed to health personnel in Norway, before it was stopped as of the occurrence of VITT. Analyzing the association of side effects with gender did not demonstrate preponderance for women but is attributed to the fact that more women than men are working in the health care sector [28]. This is in line with larger observational studies [1]. In rare cases, the vaccine may unintentionally be administered intravenously. This may cause activation of granulocytes that may be trapped in the pulmonary capillaries and cause tissue damage. Boneschansker et al. have shown that NETs from granulocytes block the microcirculation much more efficiently than thrombi [29].

In conclusion, we report an increase in NETs and its constituent, calprotectin, together with syndecan-1 in ChAdOx1-vaccinated individuals. Since NETs and calprotectin also discriminate between those with severe and milder side effects after such vaccination, these inflammatory markers could be used to confirm a more severe clinical outcome after ChAdOx1 nCov-19 vaccination. 

## Figures and Tables

**Figure 1 vaccines-10-01267-f001:**
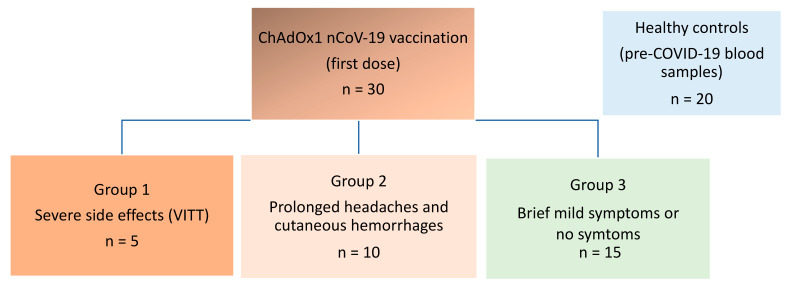
Schematic overview of groups of patients Four groups of participants were included. Group 1-3 had received ChAdOx1 nCoV-19 vaccination (first dose) and experienced complications of different levels of severity. Group 4 consisted of non-vaccinated healthy blood donor controls from pre-COVID era (2015).

**Figure 2 vaccines-10-01267-f002:**
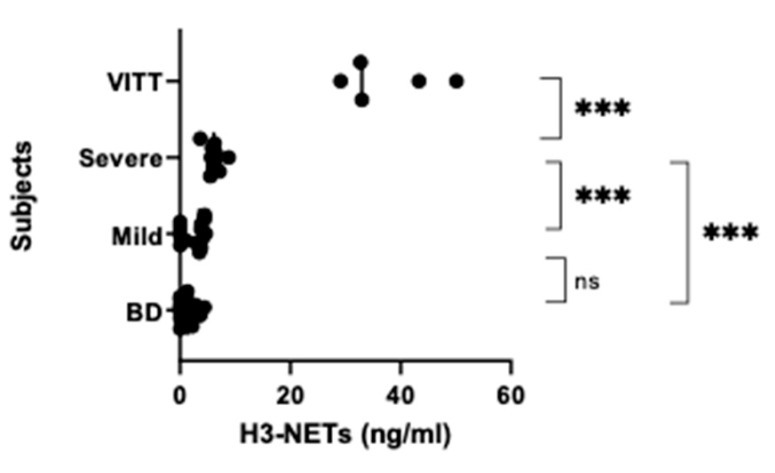
NETs levels. NETs levels were examined by a histone-3/ calprotectin ELISA in blood samples from ChAdOx1 nCov-19 vaccinated individuals that were divided into three groups (see Figure 1) according to severity of side effects. Group 1: VITT (n = 5), Group 2: Severe prolonged symptoms and signs (n = 10) who were assessed for but did not have VITT, Group 3: Mild brief symptoms (n = 15) with some headaches and general malaise who had visited emergency units but not needing further medical attention. Controls were samples from healthy (pre-COVID-19) blood donors (BD) (n = 20) donated before the pandemic. *p* < 0.0001 for one way ANOVA, for pairwise comparisons (two-tailed *t*-test): *** *p* < 0.0001, ns (not significant, here: *p* = 0.088).

**Figure 3 vaccines-10-01267-f003:**
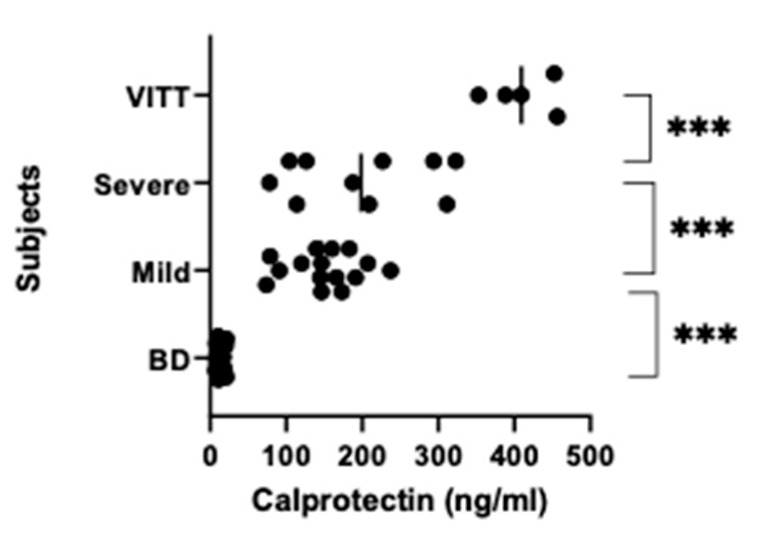
Calprotectin levels. Calprotectin levels were examined in the same samples as in Figure 2 by a mixed monoclonals (MiMo) ELISA. *p* < 0.0001 for one way ANOVA, for pairwise comparisons: *** *p* < 0.0001.

**Figure 4 vaccines-10-01267-f004:**
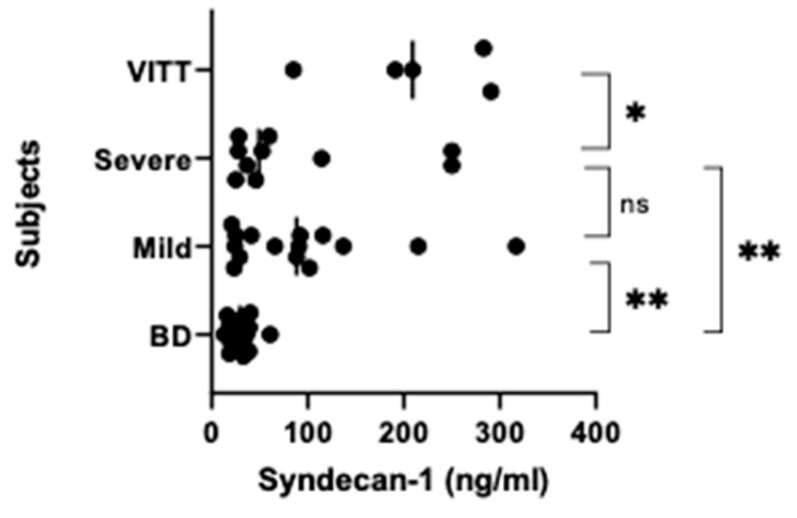
Syndecan-1 levels. Syndecan-1 levels were examined in the same samples as in Figure 2 by use of a commercial ELISA kit. *p* < 0.0001 for one way ANOVA, and for pairwise comparisons: * *p* = 0.023 and ** *p* < 0.005. ns (not significant).

**Table 1 vaccines-10-01267-t001:** Characteristics of the study participants.

–	Group 1	Group 2	Group 3	Group 4
Age (range)	39 (32–54)	36 (28–48)	38 (27–63)	45 (24–64)
Sex (% women)	80	100	80	50
Platelet count Mean (SD)	27 (17.2) *	275 (88)	303 (57)	145-390 **

Group 1: patients with VITT. Group 2: patients with prolonged symptoms of headache and skin hemorrhages, Group 3: Individuals with brief or no symptoms, Group 4: healthy controls. Data for age and platelet counts (10^9^/L) are medians and ranges and mean ± SD, respectively. For One-way ANOVA comparison of age between groups; *p* = 0.200. For One-way ANOVA comparison of platelet counts between patient groups; *p* < 0.0001. * *p* < 0.0001 vs. group 2 or 3. ** Normal reference range for blood donors is given for lack of individual values at time of sampling.

**Table 2 vaccines-10-01267-t002:** ChAdOx1-vaccinated patients with severe symptoms (VITT) (Group 1; no. 1–5), prolonged symptoms and signs (Group 2; no 6–15) or with brief mild symptoms (Group 3; no 16–30).

	Patient No	Age (Years) and Gender	Symptoms	PLT	Signs	Comments
Group 1 (VITT)	1	37 F	Fever, HA, visual disturbance	22	VITT	Fatal
	2	42 F	HA, drowsiness	14	VITT	Fatal
	3	32 M	Back pain	10	VITT	Full recovery
	4	39 F	HA, abdominal pain	70	VITT	Full recovery
	5	54 F	HA, hemiparesis	19	VITT	Fatal
Group 2 (Prolonged symptoms)	6	43 F	Mild HA (1 wk), nausea for 2d	149	Large hematoma on lower extremities	–
	7	40 F	Mild to moderate HA (3 wks), fever, fatigue, visual disturbance	420	Bruises	–
	8	32 F	Mild to moderate HA (2 wks)	386	Bruises	–
	9	32 F	Mild HA (2 wks)	-	Bruises	–
	10	32 F	Mild HA (4 wks)	267	Bruises and petechiae	–
	11	31 F	Mild to moderate HA (1 wk), muscle aches, fever	212	Petechiae	–
	12	28 F	Mild to strong HA (2 wks), fever	255	Bruises and petechiae	–
	13	45 F	Moderate to strong HA (3 wks), fever	250	Bruises and petechiae	
	14	48 F	Strong HA (3 wks), fever, vertigo, joint aches	-	Bruises and petechiae	–
	15	42 F	Moderate HA, fever, chills, nosebleed	263	Bruises and petechiae	–
Group 3 (Brief or no symptoms)	16	38 F	Fever for 2 days, tenderness in arm for 10 days	285	–	–
	17	42 F	Fever, muscle pain, influenza-like symptoms for 1 d	333	–	–
	18	44 F	Light symptoms for 2 d: tenderness in arms, fatigue, HA	201	–	–
	19	35 F	Influenza-like symptoms with HA and fatigue for 1 d	328	–	–
	20	27F	HA, dizziness, nausea for 1.5 d	291	–	–
	21	59 F	Tenderness in arm, stiffness in joints, fatigue for 1 d	240	–	–
	22	32 F	Fever for 1d, HA, back pain	207	–	–
	23	32 F	Fever for 24 h	388	–	–
	24	32 F	Fever and muscle ache 3 days	322	–	–
	25	61 F	Fever, cold sweats, some HA for 1.5 d	265	–	–
	26	50 F	High fever for 1d with nausea, then low fever for next day	373	–	–
	27	35 F	Fever and HA for 1d, tenderness in arm for 1 wk	338	–	–
	28	43 M	Fever for 1d, tired, mild HA for <1 d	297	–	–
	29	34 M	Congestion, head 1d	300	–	–
	30	63 M	No symptoms	373	–	–

Abbreviation: HA: Headache. PLT: platelet count. d: days. Wk: week.

**Table 3 vaccines-10-01267-t003:** Correlations between values for inflammatory markers in ChAdOx1-vaccinated subjects with VITT, prolonged severe or brief mild symptoms and side effects degree Pearson correlation coefficient (r).

Parameters	Side Effects	Platelet Count	H3-NETs	Calprotectin	Syndecan-1
Side effects	1	−0.763 **	0.745 **	0.902 **	0.590 **
Platelet count	–	1	−0.813 **	−0.626 **	−0.488 *
H3-NETs	–	–	1	0.818 **	0.427 *
Calprotectin	–	–	–	1	0.189 ^ns^
Syndecan-1	–	–	–	–	1

* *p* < 0.05, ** *p* < 0.001. Side effects score was 3 for VITT, 2 for prolonged serious symptoms, 1 for brief mild symptoms, and 0 for unvaccinated blood donor controls from pre-COVID era. ^ns^: not significant

## Data Availability

Not applicable.

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
