# Peer review of "Elevated NETs and Calprotectin Levels after ChAdOx1 nCoV-19 Vaccination Correlate with the Severity of Side Effects"

_vaccines, 2022, doi:10.3390/vaccines10081267_

Round 1

Reviewer 1 Report

This manuscript is an observational study of inflammatory markers correlating with severity of vaccine-induced immune thrombotic thrombocytopenia (VITT) after ChAdOx1 nCoV-19 vaccination. Although the study is an observational study with very small numbers of participants, the strong correlation between several key inflammatory markers and VITT side effects after vaccination are compelling and contribute to further understanding of the serious, albeit rare, VITT side effects that have resulted in fatalities after ChAdOx1 nCoV-19 vaccination. The paper is well-written, organized, and results presented clearly. Minor comments are listed for consideration by the authors.

1.      Introduction: How is VITT syndrome treated and how would such a regimen be implemented to increase survival rate? Could this be applied to other forms of thrombotic thrombocytopenia?

2.      What was the time between vaccination, symptoms, and blood draw for analysis?

3.      Blood samples from the healthy donors were collected in 2015, prior to the pandemic. Did the authors consider including samples from healthy donors collected during the same timeframe as the other groups of participants?

4.      Methods: Were positive and negative controls included in the various assays used? Were steps taken to establish sensitivity, specificity, and reproducibility of the assays?

5.      Table 1: Define footnotes 1 and 2. Is there a difference in the normal reference range for platelet count between male and female populations in the same age groups?

6.      Table 2: Define abbreviation (HA)

7.      Figures 2 – 4: Suggest adding Group number to each of the groups listed on the Y-axis.

Author Response

Reviewer 1

This manuscript is an observational study of inflammatory markers correlating with severity of vaccine-induced immune thrombotic thrombocytopenia (VITT) after ChAdOx1 nCoV-19 vaccination. Although the study is an observational study with very small numbers of participants, the strong correlation between several key inflammatory markers and VITT side effects after vaccination are compelling and contribute to further understanding of the serious, albeit rare, VITT side effects that have resulted in fatalities after ChAdOx1 nCoV-19 vaccination. The paper is well-written, organized, and results presented clearly. Minor comments are listed for consideration by the authors.

  1. Introduction: How is VITT syndrome treated and how would such a regimen be implemented to increase survival rate? Could this be applied to other forms of thrombotic thrombocytopenia?

Response: IVIG and anticoagulation are cornerstones of VITT treatment. The implementation of this treatment has improved survival rates greatly (Pavord et al., NEJM 2021). VITT is a distinct form of thrombotic thrombocytopenia, different from TTP and HIT, therefore, treatment is different. However, plasma exchange and withdrawal of heparin has been suggested forms of therapy, but evidence is sparse.

Change in manuscript: We have added two sentences about treatment in the introduction section (page 1, lines 39-40).

  1. What was the time between vaccination, symptoms, and blood draw for analysis?

Response: Thank you for addressing this. issue This essential information has unfortunately been lost in the revision of the manuscript. For the VITT patients (Group 1) the time between vaccination and symptoms was 7-10 days and blood draw was performed at admittance- 10-14 days post vaccination. The vaccines with prolonged side effects Group 2 had onset at a median of 1.5 days after vaccination and lasted for a median of 2.8 weeks and median number of days from vaccination to blood sampling were 26 (range 10 to 37). The onset of symptoms in vaccinees with brief or no symptoms (Group 3) was after 0-2 days.

Change in manuscript: The time from vaccination to blood sampling is already described in the Methods section (page 3, lines 107-108). The information about onset and duration of symptoms has also been added (page 2-3, lines 93-98).

  1. Blood samples from the healthy donors were collected in 2015, prior to the pandemic. Did the authors consider including samples from healthy donors collected during the same timeframe as the other groups of participants?

Response: Thank you for this question. We did consider including healthy donors at the time of our study, but because such a large proportion of the blood donors had either been vaccinated for Covid-19 or had undergone infection, we considered historical data just as relevant for this study.

Change in manuscript: No changes

  1. Methods: Were positive and negative controls included in the various assays used? Were steps taken to establish sensitivity, specificity, and reproducibility of the assays?

      Response: Positive and negative controls were included in the tests. Questions related to sensitivity, specificity and reproduction of the assays (NETS and calprotectin assays) have been met in the method papers of these assays which are referred to in the current paper. The Syndecan-1 assay is a commercial assay, and we refer to information from the manufacturers (Diaclone, 950.640.192, Besançon, France)

      Change in manuscript: No changes made

  1. 5.Table 1: Define footnotes 1 and 2. Is there a difference in the normal reference range for platelet count between male and female populations in the same age groups?

      Response: Regarding the footnotes; thank you for pointing this out. These footnote numbers should have been removed before submission. The reference range for platelet count is higher for females than males in the same age groups, we used reference ranges in Norway in the current paper.

      Change in manuscript: Footnotes 1 and 2 have been removed in Table 1

  1. Table 2: Define abbreviation (HA)

      Response: We agree. HA is the abbreviation for headache.

      Change in manuscript: Abbreviation definition is added in the footnote of Table 2.

  1. Figures 2 – 4: Suggest adding Group number to each of the groups listed on the Y-axis.

Response: We agree that this may improve the figures. Unfortunately, this is not possible within the time we have for revising the manuscript. 

 Change in manuscript: No changes made

Reviewer 2 Report

The manuscript titled " Elevated NETS and Calprotectin levels after ChAdOx1 nCoV-19 vaccination correlate with the severity of side effects" is a well-written manuscript on potential early detection markers for developing severe side effects to the ChAdOx vaccine. The background and methods are well-written and detailed. Figures are easy to read. And the discussion is well organized and strongly supports the findings of the work. Great manuscript! Extremely valuable information for combating side effects associated with COVID 19 vaccination. 

Author Response

Thank you very much for your comments on our manuscript.